# A Situational Analysis of Attitudes toward Stray Cats and Preferences and Priorities for Their Management

**DOI:** 10.3390/ani14202953

**Published:** 2024-10-14

**Authors:** Jacquie Rand, Rebekah Scotney, Ann Enright, Andrea Hayward, Pauleen Bennett, John Morton

**Affiliations:** 1Australian Pet Welfare Foundation, Kenmore, QLD 4069, Australia; aenright7@gmail.com (A.E.); dolly5664@gmail.com (A.H.); 2Faculty of Science, School of Veterinary Science, The University of Queensland, Gatton Campus, QLD 4343, Australia; rebekah.scotney@uq.edu.au (R.S.); j.morton@uq.edu.au (J.M.); 3School of Psychology and Public Health, La Trobe University, Bendigo, VIC 3552, Australia; pauleen.bennett@latrobe.edu.au; 4Jemora Pty Ltd., P.O. Box 5010, East Geelong, VIC 3219, Australia

**Keywords:** cat management preferences, sterilizing, euthanizing, kittens, one welfare, wildlife

## Abstract

**Simple Summary:**

Australia’s current strategies for urban cat management are outdated and ineffective, and are typically based on mandated containment and “trap, adopt, or euthanize” responses for nuisance cats. We undertook a situational analysis before implementing a cat management program based on free sterilization for owned, semi-owned, and unowned cats in a targeted area with a high per capita cat intake into the receiving shelters. Before implementing the intervention, we aimed to understand the attitudes and behaviors towards wandering and stray cats, as well as community preferences and priorities for their management. Stray or wandering cats were observed by many respondents (71%), primarily at private residences and in alleyways or streets, which caused serious or moderately serious problems for 38% of respondents who saw stray or wandering cats. Similar levels of concern were expressed in regards to the stray cats killing native birds, killing native animals, creating noise, and soiling. The respondents preferred sterilization over euthanasia for managing stray cats. Only a minority of respondents were satisfied with current local council cat management. Increasing public awareness about the advantages of cat management based on sterilization could further increase community support for this approach.

**Abstract:**

Current cat management approaches are outdated and ineffective, failing to reduce stray cat numbers or related complaints and negatively impacting the job satisfaction and mental health of veterinary, shelter, and municipal staff. We undertook a situational analysis prior to implementing a Community Cat Program based on free sterilization of owned, semi-owned, and unowned cats in the city of Ipswich, Queensland, Australia. The study involved 343 residents in three suburbs in Ipswich, Queensland, Australia with high per capita intake of cats into the receiving shelter and municipal pound. We investigated the prevalence and impacts of free-roaming cats in urban areas, focusing on sightings, associated issues, and community preferences for cat management. Stray cats were observed by many respondents (71%), primarily at private residences (52%) and in alleyways or streets (22%), which caused serious or moderately serious problems for 38% of those who saw stray or wandering cats. Key concerns included the killing of native birds (38%) and animals (35%), noise (33%), and soiling (32%). Actions taken by respondents who saw stray or wandering cats included chasing them away or using deterrents (25%), capturing the cat for removal or calling council (18%) and preventing home entry (14%). Respondents’ priorities for the local government management of cats included preventing kittens from being born (94% of respondents) and stopping cats from preying on native animals (91%); reducing disease spread to pets (89%), wildlife (89%), and humans (87%); decreasing stray cat numbers (75%); and preventing cat fights (70%). Respondents favored sterilization (65%) over euthanasia (35%), aligning with the results of previous research. Cat ownership and feeding unowned cats were predictors of management preferences. Only 29% of respondents were satisfied with the current local council management of the problem. Information on the benefits of management by sterilization could further enhance community support.

## 1. Introduction

In Australia, management strategies for free-roaming cats in urban areas have been historically ineffective [1,2]. Current cat management programs—many of which are decades old—are based on depopulation methods consisting of trapping free-roaming and nuisance cats in an effort to control their numbers. These strategies have not successfully reduced the number of cat-related complaints, nor the number of cats impounded and euthanized [1,2]. This negatively impacts the mental health of staff involved in their euthanasia, as well as people caring for the cats [2,3,4,5,6,7,8,9]. On average across Australia in 2018–2019, approximately 33% of cats entering shelters and pounds were euthanized, with an estimated 50,000 cats euthanized in Australia annually [1]. The majority of these are healthy stray cats and less than one year old.

In contrast, cat population management programs that focus on sterilizing semi-owned and unowned cats, commonly referred to as trap–neuter–return (TNR) programs, are effective in reducing cat-related complaints and shelter intake [10,11,12,13], provided that they are of sufficient intensity and duration. However, these programs are currently illegal in Australia under various legislation related to biosecurity, abandonment, and mandated cat containment, and there is considerable opposition to them in Australia, mainly because of wildlife concerns [14,15,16,17,18].

The intrinsic predatory nature of cats has drawn negative attention from conservationists and wildlife carers, generating public concern and debate regarding cat management. Frequent media releases in Australia emphasizing the risk of pet cats to wildlife contribute to this perception and are part of campaigns to gain social license for cat management [19]. Local governments are increasingly implementing stricter requirements for pet cat confinement (e.g., leash laws in USA) to address these concerns [20,21,22,23,24]. However, these measures fail to address the reality that most stray cats are unidentified, owned, or semi-owned cats (cared for by people who do not perceive they are the owner) in low socioeconomic areas, where residents typically lack the financial resources to build cat containment systems and in which rental properties are often unsuitable [25]. These laws and bylaws can create barriers to the ownership, sterilization, and microchipping of owned and stray cats, hence perpetuating cat overpopulation. International and Australian scientific evidence repeatedly demonstrates that targeted, intensive sterilization programs will significantly reduce the number of stray cats that are impounded and euthanized by municipal authorities and shelters, and also significantly reduce cat-related calls to local government authorities [2,11,12,26,27,28,29,30,31,32,33,34,35].

The Australian Pet Welfare Foundation (APWF) secured the necessary research permits from the Queensland (QLD) Department of Agriculture and Fisheries (DAF) to conduct a research trial of a free cat sterilization and microchipping program for semi-owned and unowned cats in the city of Ipswich, Queensland. This allowed us to evaluate its efficacy in tackling the root causes of cat overpopulation and associated nuisance issues.

Effectively managing urban stray cats requires a deep understanding of community attitudes and cat-caring behaviors. Conducting a situational analysis prior to implementing interventions is vital, as it establishes a baseline for evaluation. Therefore, before implementing the targeted, high-intensity sterilization program for owned, semi-owned, and unowned cats, a situational analysis was performed to obtain baseline data on cat ownership and sterilization rates and issues associated with free-roaming urban cats, as well as community priorities for their management.

The first part of the situational analysis relating to cat ownership and caring behaviors found that 75% of respondents owned a pet, with 35% owning cats and 53% owning dogs [36]. At least 3–4% of respondents fed cats they did not know to be owned, and therefore, they were considered semi-owners. Most respondents strongly agreed with liking dogs (76%) and native birds and animals (73%). Only 41% felt the same for cats, and 18% strongly disagreed with liking cats, whereas few (2%) strongly disagreed with liking dogs. Men and women similarly liked dogs and wildlife, whereas fewer men strongly agreed with liking cats (31% versus 46%).

Previous Australian research shows strong community support for non-lethal approaches to managing stray cats, with 78% of Brisbane respondents preferring sterilization-based population control methods over euthanasia when provided additional information on the benefits of sterilizing over culling [37]. However, significant knowledge gaps remain. Further research is needed to understand community attitudes towards stray cats and their management in urban and peri-urban areas in Australia. This current study aimed to identify the nature of the cat-related concerns of local residents and to understand their preferences and priorities regarding the outcomes of cat management. Such information has the potential to enable authorities and key stakeholders to efficiently plan and communicate key messaging important to the community in order to increase support for new cat management strategies. This could also facilitate the efficient use of resources by increasing community engagement.

The study aim was to conduct a situational analysis before implementing a Community Cat Program based on targeted free sterilization and microchipping. One part of the situational analysis, focusing on cat ownership and cat-caring behaviors, is published [36]. This current paper reports the outcome of the second part of the situational analysis, which aimed to explore community attitudes towards stray cats, management priorities, preferences for management, and predictors of these preferences and to provide a basis for a subsequent impact analysis. The objective of the overall project is to better inform the public of strategies and legislative changes that would improve the effectiveness of urban cat management. This would enhance outcomes for stray and surrendered cats, mitigate issues associated with free-roaming cats, and reduce mental health impacts associated with euthanizing healthy and treatable cats and kittens on caregivers, veterinarians, animal management officers and shelter staff.

## 2. Materials and Methods

A survey was undertaken using online questionnaire software provided by ArcGIS Survey123 [38]. Respondents were requested to provide information on their gender, suburb, age, education, culture, pet ownership, their interactions with different types of animals, and their beliefs about stray cat management (Table A1). Results regarding pet ownership; cat-caring behaviors; whether respondents liked dogs, cats, and native birds and animals; and whether they fed owned or unowned animals, including dogs, cats, and wildlife, are reported in a preceding publication [36]. This report focuses on perceived issues with free-roaming cats, preferences for their management, and priorities for outcomes of that management.

Data were obtained from people in three suburbs (Goodna, Rosewood, and Redbank Plains), with populations of 10,391, 3263, and 24,349, respectively [39,40,41], in the city of Ipswich, Queensland. These suburbs were selected for the implementation of a pilot Community Cat Program based on the free sterilization and microchipping of owned, semi-owned, and stray cats because these suburbs had a higher per capita intake of cats into the receiving shelter and municipal (council) pound. They also exhibited lower scores for socioeconomic advantage and disadvantage than the averages for Queensland and Australia, as well as higher levels of unemployment and lower than average levels of education [36,39,40,41]. The majority of respondents were women (66% compared to 34% men), and women were more likely than men to own cats and/or dogs.

The questionnaire (Table A1) was administered from 6 June 2020 to 25 September 2021, initially face-to-face (31 respondents) and then by telephone (330 respondents) due to restrictions associated with the COVID-19 pandemic. Respondents to the door-knock or telephone call who were 18 years or older and Australian residents were invited to participate anonymously in the survey. Whoever answered the door or phone, if eligible, was asked to complete the survey. During the consent process, potential respondents were informed that the “purpose of this survey is to gauge community attitudes towards pet ownership and the management of unowned and stray cats and kittens living in the city and suburbs”.

The door-knock routes used were those previously chosen for walking transects for counting cats, and along those routes, any house where the front door was accessible, where it appeared safe to enter the property, was approached. Of the 31 residences recorded as being approached, responses from one person from each of 28 residences were obtained. For the remaining three residences, no respondent was enrolled for the following reasons: the householder declined; the person available was aged under 18 years and thus, was ineligible; or no one was home (one residence each).

Landline and mobile telephone numbers for the three suburbs were obtained from Sample Pages (Cremorne, VIC, Australia) for 6004 phone numbers (2252 landline; 3752 mobile), which were washed to remove people on the do not call list, and “may also have contained numbers seeded in the list by Sample Pages to ensure that the numbers were being used in accordance with the agreement” (which specified the duration for which they could be used). Of these 6004 phone numbers, 3190 phone numbers were randomly chosen to be called, of which 2169 (68%) were called once, 766 (24%) were called twice or more, and 255 (8%) were called three times or more. Of those, 1116 (35%) calls went straight to voice mail, and 383 (12%) were not answered, resulting in 1499 (47%) people not being reached for communication. Of the remaining 1691 (53%) potential participants who answered the calls, 240 (14.2%) were no longer living in the area; 71 (4.2%) of calls ended immediately; 3 (0.2%) were busy but happy to reschedule, but no one got back to them; and 1047 (61.9%) declined further communication, resulting in 330 (19.5%) participants who answered the phone and who agreed to participate in the survey; of these, 315 eligible responses were provided.

Therefore, of the 3190 phone numbers called, 1380 (1691-240-71) answered the phone and were known to live in the area; of these, 330 residents agreed to participate, and 315 provided an eligible response via phone (the response rate of those who answered the phone and lived in the area was 315/1380 = 22.8%). However, the response rate for all phone numbers called (3190) was only 11.5%, if 14.2% of these 3190 numbers were to people who no longer lived in the area.

Thus, data were entered for a total of 361 residences (31 in person, 330 by phone) and for 343 of these (28 and 315, respectively), an eligible respondent provided responses. Of those who specified a residential postcode (*n* = 320), 2.5% (8) gave a postcode not associated with the three targeted suburbs. All eight were retained in the study because prior to participating in the survey, it was confirmed that the person did reside in the nominated area. Hence, it was assumed the postcode was recorded incorrectly when the questionnaire was administered. Of the 343 respondents, the questionnaire was completed in full by 96% of all respondents (330/343), with 78% (22/28) of the face-to-face and 98% (308/315) of the telephone interviews fully completed, demonstrating a good level of engagement. The relative proportions of eligible responses obtained from residents in the three key suburbs were in approximate proportion to their relative populations.

The city of Ipswich bylaws (ordinances) require that cats be contained within their owner’s property, and a permit must be obtained for keeping three or more cats. Respondents were asked whether they observed stray or wandering cats in their neighborhood, the places they were seen, and whether they caused a problem for respondents, which was assessed using a four-point Likert scale (serious problem, moderate problem, mild problem, no problem) (Table A1). Respondents were asked why stray cats were a problem for them and what actions they had taken against stray or wandering cats.

Respondents’ satisfaction level about the local management of stray and wandering cats was assessed using a six-point Likert scale (extremely satisfied, somewhat satisfied, neither satisfied nor dissatisfied, somewhat dissatisfied, extremely dissatisfied, not sure). Their beliefs about how stray cats should be managed (euthanizing, sterilizing, or just leaving the cats alone) were assessed using a five-point Likert scale (extremely supportive, somewhat supportive, neither supportive nor unsupportive, somewhat unsupportive, extremely unsupportive). Respondents’ attitudes and concerns about the elements important to them in regards to their councils’ decisions concerning the best type of program to manage unowned stray cats in their suburbs (11 items) were assessed with a five-point Likert scale (very important, important, moderately important, low importance, not at all important.

Respondents were also asked if they had heard about the Australian Pet Welfare Foundation’s Community Cat Program for the sterilization of cats in their areas. This question was added because the survey data were originally intended to be collected before the start of the sterilization program. However, the COVID outbreak stopped face-to-face data collection, necessitating modifications to the survey data to be collected by telephone; consequently, some of the data collected by telephone were obtained after the beginning of the sterilization program. Therefore, we conducted an analysis to determine whether knowledge of the sterilization program influenced respondents’ choices for support of either lethal or non-lethal cat management programs.

The denominators for the reported results varied due to non-responses to some questions and inconsistent responses to different questions by individual respondents. In addition, some analyses were for specific subsets of the 343 respondents. Each respondent was able to list multiple location types where stray or wandering cats were seen. For each location type, we calculated percentages of respondents who listed that location type. We also calculated the number of location types listed by each respondent (where each respondent–location type combination constituted one “location”), summed those over all respondents, and calculated percentages of the total number of locations for each location type. The problems caused by stray or wandering cats and the actions taken against stray or wandering cats were analyzed using the same approach.

The proportions were compared using two-sided exact *p*-values, calculated using Fisher’s tests in Stata (version 18, StataCorp, College Station, TX, USA). The study was approved by the University of Queensland Human Ethics Committee (approval number 2014000597).

## 3. Results

### 3.1. Free-Roaming Cats and Associated Issues

Most respondents (71%, or 238/337) reported seeing stray or wandering cats in their city or town. The majority saw them at private residences (89%), and private residences constituted 53% of reported locations where stray cats were seen (Table 1). Fewer respondents who saw stray or wandering cats observed them in alleyways or streets (38%), and alleyways or streets constituted the next most frequently reported location (23%), followed by suburban parks (9%). Stray cats were observed in a wide variety of other locations, including near food outlets and other commercial businesses, vacant blocks or buildings, train stations, and drains.

Of the 238 respondents who saw stray or wandering cats, 74% (176) indicated that stray cats were a problem for them, indicating at least one reason for this. Reasons included the cats killing native birds (38% of respondents who saw stray or wandering cats); killing small native animals (35%); causing a nuisance through noise and fighting (33%), defecating, and urinating (32%); and attacking their cat or dog (15%) (Table A2). Of lesser concern were stray cats entering (and damaging) the respondent’s property (8%) and the welfare of the stray cats (4%). Responses were generally similar by gender (Table A3).

The respondents indicated the extent to which stray or wandering cats were a problem to them in each respondent location. The cats were problematic (a serious or moderate problem) in the locations where the respondents lived for 38% of respondents and in local public places for 17% of respondents. Fewer respondents reported that the cats were a serious or moderate problem in the locations where the respondents worked, studied, or spent most of their time away from home (11%) (Table 2).

Of the 238 respondents who reported seeing stray or wandering cats, 25% chased the cat(s) away or used a deterrent, and 14% took action to prevent the cat(s) from entering their homes (Table 3). Some respondents (8%) captured the cat(s) for removal or contacted the council, who loaned out cat traps (8%), and 5% provided the cat(s) with food. Of the 238 respondents who reported seeing stray or wandering cats, 52% (123) reported that they took no action.

### 3.2. Preferences Regarding Stray Cat Management Methods

Respondents were asked, “If two methods of cat management were equally effective in decreasing unowned stray cat numbers over time and the problems they cause, how strongly would you support or approve of the two methods of control for healthy cats that cannot be readily adopted?” They were first given the option of “catching and humanely euthanizing unowned stray cats that cannot be readily adopted”, and 56% were supportive, 30% were unsupportive, and 15% were neither supportive nor unsupportive (Table 4). Respondents were then asked for their opinion on “catching, sterilizing, vaccinating and returning healthy stray cats to where they live”, with 52% supportive, 33% unsupportive, and 15% neither supportive nor unsupportive. Thirdly, respondents were then asked, “How strongly do you support just leaving the unowned stray cats alone?”, with most (69%) somewhat or extremely unsupportive (Table 4), with similar lack of support from women and men. Respondents could be supportive of more than one of these methods or none of these methods.

For the last 3 months of the data collection period (i.e., from June 2021 on), respondents were also asked, “If studies showed, that to decrease stray cat numbers in Ipswich, 3000 stray cats would need to be either killed or sterilized, which method of stray cat management would you prefer?” and were required to select one of these two options. The majority of respondents preferred sterilizing cats (65%, or 184/282) compared to 35% preferring euthanizing as a method for stray cat management. A higher proportion of women (75%, or 141/189) compared to men (46% or 43/93) supported sterilizing cats; thus, a higher proportion of men supported euthanizing cats (54%) compared to women (25%). For those respondents in the younger than 60 years age bracket, a higher proportion supported sterilizing rather than euthanizing (74%, or 133/179) compared to results for the respondents in the 60 years or older age bracket (49%, or 49/100, supported sterilizing rather than euthanizing).

Most respondents (77%, or 257/334) had not heard of the Australian Pet Welfare Foundation’s Community Cat Program (a program that would sterilize cats in their area for free). Respondents who had heard about the program were more likely to support sterilizing (79%, or 45/57) compared to those that had not heard of the program (62%, or 139/224; *p* = 0.02).

### 3.3. Associations between Respondents Fostering Wildlife and Those Who Owned Birds, Dogs, Cats, or Other Pets and Preferences for Stray Cat Management

When questions about the choice of management were asked sequentially, the proportions of respondents supporting euthanizing unowned stray cats were similar for those who did and those who did not foster wildlife; feed unowned birds, possums, or cats; or own birds or dogs (Table A4). However, fewer respondents who owned cats were supportive of euthanasia (47%) compared to those who did not own a cat (61%; *p* = 0.02) (Table A4). Similarly, more respondents who fed unowned cats supported sterilizing stray cats (91%) compared to those who did not feed strays (51%). Support for leaving stray cats alone was similar for those respondents who did and those who did not foster wildlife; feed unowned birds, possums, or cats; or own birds, dogs, or cats (Table A4).

When respondents were asked to choose which of two methods they preferred for equally decreasing stray cats (killing or sterilizing), the proportions of respondents supporting catching, sterilizing, vaccinating, and returning the healthy stray cats to where they live were similar for those who did and those who did not foster wildlife; feed unowned birds, possums, or cats; or own birds or dogs (Table 5). However, a greater number of the respondents who owned cats were supportive of sterilizing (75%) compared to the number of those that did not own cats (60%; *p* = 0.01).

### 3.4. Associations between Respondents Liking Dogs, Cats, and Native Wildlife and Preferences for Stray Cat Management

Associations between respondents liking dogs, cats, and wildlife and preferences for stray cat management are shown in Table A5. There was some evidence that respondents who liked cats were more likely to prefer sterilizing stray cats instead of euthanasia.

### 3.5. Issues of Importance to Respondents Regarding Council Stray Cat Management

Only 29% of respondents were extremely or somewhat satisfied with the way stray cats with no known owner were being managed in their town (Table A6). Notably, 18% of respondents were dissatisfied to some degree regarding stray cat management, with an additional 14% remaining neutral on the issue and a very high percentage (39%) stating that they were unsure. Slightly more residents (35%) were satisfied with how wandering cats with a known owner were managed. 

The features of a management program regarded by the greatest proportion of respondents as important or very important were stopping stray kittens from being born (94%); decreasing the killing of birds and small native animals by cats (91%); and reducing the risk of the spread of disease to pets (89%), wildlife (89%), and humans (87%) (Table 6).

Most respondents also considered decreasing stray cat numbers in their suburb to be important or very important (74%). Over half of the respondents considered removing the cats causing concern to them as important or very important (61%), with 17% considering this to be moderately important. In contrast, fewer respondents considered the cost-effectiveness of the programs for councils and ratepayers to be important, with 59% indicating that it was important or very important and 28% considering cost-effectiveness to be moderately important.

## 4. Discussion

In Australia, management strategies for free-roaming urban cats remain largely ineffective, primarily focusing on outdated depopulation methods that have not successfully reduced cat-related complaints or the number of impounded and euthanized cats [1,4]. This study’s situational analysis in Ipswich, Queensland, highlights key findings that underscore the inadequacy of current practices. Only a minority of respondents were satisfied with the local government’s management of stray and wandering cats, and a significant majority reported frequent sightings of stray cats, primarily at private residences, and expressed concerns about wildlife predation and nuisance behaviors. Importantly, the data revealed robust community support for non-lethal, sterilization-based management programs.

### 4.1. Free-Roaming Cats, Associated Issues, and Actions Taken against Them

In our study, stray and wandering cats were common and were observed by nearly three-quarters of respondents. In a study from the adjacent city of Brisbane, private residences were the most common location for reported stray cat sighting (21%), followed by alleyways (15%) and commercial businesses (15%) [33,37]. These results were similar to those of an Australia-wide study in which multi-cat sites (colonies) were most commonly situated at private residences (26%), followed by industrial areas or factory complexes (20%) and alleyways (13%) [42]. In our study area, industrial areas and factory complexes were an uncommon feature of the urban landscape, and most cats were seen at private residences, followed by alleyways.

Free-roaming cats were most frequently seen in locations where respondents lived, with over one-third stating that the cats caused a serious or moderately serious problem for them. In the 2019 Brisbane study, concerns for wildlife were common (killing birds, 38%, and small animals, 39%), but a greater proportion of people were concerned with wandering or stray cats fighting (49%) and soiling (48%) [37]. In our study, similar levels of concern were expressed for wildlife predation and nuisance issues, although the greatest proportion of respondents were concerned about the killing of native birds. In Japan, the nuisance created by the smell of feces and urine was a strong concern for 30% of respondents [43], while the results from a pilot study conducted in India showed that most respondents reported little concern regarding soiling [44]. In contrast with our results, a Bulgarian study reported that 82% of respondents felt sorry for stray cats and did not believe stray cats were a nuisance [45].

Cats’ inherent predatory instincts have drawn criticism from conservationists and wildlife caretakers, sparking public concern and debate over cat management and containment strategies. It is interesting to note that the predation of native animals was the highest concern of the respondents, but it is unlikely that predation events would be as frequently experienced as nuisance behaviors. However, in Australia, frequent media releases highlight the threat pet cats pose to wildlife as part of efforts to secure public approval for the lethal management of feral cats and the acceptance of mandatory containment policies for pet cats [19]. Unfortunately, these mandates make cat ownership illegal for people unable to contain their cats and contribute to the ongoing cycle of unwanted litters of kitten being born. This is because those who cannot contain their cats are unable to take ownership and therefore, cannot have the animals sterilized because it is illegal for veterinarians in Queensland to sterilize an unowned cat without a restricted matter permit [46].

In Queensland, local governments are responsible for domestic animal management and are accountable for addressing nuisance cat complaints. The city of Ipswich provides trap cages for complainants to trap and impound trespassing cats and will also trap and impound cats, if required. Notably, 18% of respondents either captured the stray or wandering cat(s) to have them removed or contacted the council. This approach incurs costs associated with the animal management officers’ time, as well as charges incurred from a service provider for holding cats for the minimal holding period (3 working days in Queensland). A recent study documented reduced complaints and costs to the council by implementing an assistive approach to help residents better care for cats when compared with those incurred using the decades-old compliance-based approach of trapping and impounding cats [2]. Many of these cats are timid and shy and are at high risk of euthanasia in a shelter or local government facility (pound) [1]. Implementing an assistive approach to urban cat management would allow councils to work with the community to reduce the number of free-roaming cats, decreasing the number of nuisance complaints and reducing their impact on local wildlife. Additionally, this would improve cat welfare and the mental health of shelter and veterinary staff, while decreasing the workload for animal management officers, increasing job satisfaction, and lowering costs for local councils. Therefore, we recommend providing information to the community on the benefits of implementing an assistive-centered approach to urban cat management. This would improve community engagement, which would ultimately enhance the wellbeing of animals, people, and the environment, consistent and in alignment with the One Welfare philosophy [47].

### 4.2. Preferences for Cat Management

When given a choice, most respondents preferred sterilization (65%) compared to killing (35%) as a method for stray cat management. This preference for non-lethal rather than lethal management aligns with prior research from a previous study in Australia showing that 78% of respondents preferred sterilization, as well as with the results of another study from Belgium showing that 90% of respondents favored non-lethal methods of cat management [37,48]. However, our study showed lower support than that reported elsewhere, likely because we provided the respondents with no further information on the negative outcomes or benefits of either method. An Australian study highlighted the importance of providing further information to respondents about the benefits and disadvantages of various cat management options, e.g., firstly, catching, sterilizing, microchipping and vaccinating stray cats, adopting them, if possible, returning healthy cats to where they were found, and euthanizing sick cats; secondly, continuing the present council culling program; and thirdly, leaving the cats alone [37]. Most respondents (68%) supported sterilizing cats, 28% supported culling, and 2% supported leaving them alone. However, the respondents were then asked, “Please consider the following findings from recent research on urban stray cats: The number of urban stray cats can be reduced by killing them, or by sterilizing them so that they are unable to have more kittens; To effectively decrease stray cat numbers by killing means that 40% of the population must be killed every 6 months for at least 10 years; In North American and Europe, sterilizing, adopting friendly cats to new homes, and returning the others to where they were found, reduces euthanasia of cats and kittens in shelters and pounds, and reduces cat-related complaints; And over time, it reduces the number of stray cats in cities at a similar rate as killing cats. Sterilizing and adopting or returning stray cats is often funded by community and welfare agencies, reducing costs to government compared to killing cats; Most urban stray cats are as healthy as owned domestic cats, and less than one in a hundred stray cats (1%) are too unhealthy to be returned to where found. Knowing these research findings, what would be your preference now for managing urban stray cats in Brisbane?” After being provided with this information, the respondents increased their support for the sterilizing cats to 78%, decreased their support for culling to 18%, and increased their support for leaving strays alone to 3%. These results show that non-lethal options for stray cat management are supported by communities, and this support can be increased by providing public information on the long-term benefits of such methods. Information should also be included on the benefits to the mental wellbeing of shelter and pound workers from the decreased euthanasia of healthy and treatable cats and kittens. The One Welfare concept emphasizes this interconnectedness between humans and animals [47].

### 4.3. Predictors of Preferences for Cat Management

#### 4.3.1. Predictors Based on Demographics

The association between gender and cat management preference in our study is consistent with the results of previous research reporting male gender being associated with a greater preference for killing rather than sterilization programs. Of concern, 54% of men preferred killing cats, even when informed that sterilizing the same number would equally decrease stray cat numbers. In contrast, 75% of women preferred sterilizing cats. These results are very similar to those from Brisbane, where more women (76%) supported non-lethal population control methods compared to men (45%) [37]. Women have been reported as being more sympathetic to unowned stray cats than men and less supportive of killing unowned stray cats [37,49,50,51,52,53,54]. However, cat owners were more supportive of sterilization, and more women in our study were cat owners than men, likely influencing the observed gender imbalance towards sterilization. Previous research has demonstrated that women show greater concern toward the welfare of animals compared to men and are more emotionally distressed by the unnecessary killing of animals [55,56,57].

For those respondents in the age bracket younger than 60 years old, a higher proportion showed support for sterilization compared to older respondents, who displayed higher levels of support for killing. The association between age and cat management preference aligns with previous research findings that found that younger individuals tend to favor non-lethal methods, such as sterilization, over lethal methods for managing animal populations [58]. These trends highlight the importance of considering demographics and priorities for stray cat management when communicating and implementing effective cat management programs which are supported by the community [59]. Generational differences may be explained by changing attitudes toward animal welfare and the treatment of animals over time. Some research shows that those in younger age brackets show greater pro-animal welfare attitudes compared to those in older age brackets [56]. In other studies, however, older age was associated with more support for non-lethal methods, and in other studies there was no association between age and preference for different cat management scenarios (killing or sterilization) [60,61], suggesting that age may not be a reliable predictor for stray cat management preference.

#### 4.3.2. Predictors Based on Pet Ownership or Caring for Unowned Animals

When the questions were asked sequentially for preferences for killing compared to sterilizing or leaving the cats alone, there were few predictors of support based on pet ownership or whether the respondents cared for unowned animals. The only predictors were regarding people who cared for stray cats (semi-owners) or were cat owners, with cat owners being significantly more likely to choose sterilization (91%). Surprisingly, there were no significant differences in support for any of the methods between those who did or did not foster wildlife, feed wild birds or possums, or who owned birds.

When respondents had to choose between euthanizing or sterilizing, the only predictor was whether or not they owned cats. This is consistent with prior literature regarding preferences for stray cat management, which indicated that cat owners were more supportive of sterilization programs compared to non-cat owners [43,52,62,63,64]. In our study, there was no difference in the level of support for either method between those who did or did not feed wildlife. Our findings underscore the influence of pet ownership on attitudes toward humane animal management methods, while highlighting that wildlife feeding behaviors may not significantly alter these views [65].

### 4.4. Important Elements of Council Stray Cat Management

Only 29% of respondents were satisfied with the local council’s management actions for stray cats, and the majority were dissatisfied, unsure, or neutral, indicating that many respondents were unfamiliar or dissatisfied with council actions for cat management. This is despite the council’s substantial outlay for cat and dog management, evidenced by council-employed animal management staff for urban animal control, and a contract with a service provider for cat and dog impoundment services in effect for up to 5 years [66].

Most respondents indicated that their highest priorities for the local government management of cats were stopping kittens from being born; decreasing the killing of birds and small animals; and reducing the spread of disease to pets, wildlife, and humans, with 88% to 94% indicating that these issues were important or very important. Of lesser importance were decreasing the number of stray cats and stopping nuisance behaviors. Cost-effectiveness for councils and ratepayers was a lower priority. These findings highlight that to gain public support for various cat management options, communication needs to include information about the effectiveness of stopping kittens being born, the fact that fewer free roaming cats reduces the threat to wildlife from predation, and that fewer unsterilized cats and fewer kittens means less disease risk to pets, humans, and wildlife [67,68]. Sterilizing cats reduces the incidence of feline immunodeficiency virus (FIV) by decreasing fighting and roaming behaviors [69,70]. Reducing the number of kittens and cats under one year of age decreases environmental contamination with toxoplasmosis oocysts, as younger cats are the primary shedders of these oocysts [67,68,71,72]. Sterilizing cats will markedly reduce the disturbing noises associated with fighting and the yowling of female cats in heat [73].

We did not include a question regarding the impact on staff of killing healthy cats and kittens, but such a question should be included in future questionnaires because the public should know that this is a consequence of current urban cat management. Euthanizing animals can significantly impact the mental health of veterinary professionals, contributing to occupational stress, compassion fatigue, and post-traumatic stress disorder (PTSD) [5,74,75,76]. In the USA, the suicide rate for animal protective service workers (shelter and animal control) is the same as for first responders, including firefighters and police officers [9], which is attributed to the emotional toll of their work. In addition to suicide, these workers also present high levels of PTSD, depression, and secondary traumatic stress [76,77]. This impact should be considered when making decisions regarding methods of animal management. We recommend engaging with the public to provide them with information on the beneficial outcomes of an assistive cat management strategy, focusing information on their highest cat management priorities such as reducing kittens being born, predation on wildlife and spread of disease, while also including information on the negative impacts on staff when they are required to kill healthy cats and kittens. By engaging with the community, support for this strategy can be further improved, which is essential to the success of a Community Cat Program [78].

### 4.5. Limitations of the Study

This study’s design has several limitations. Firstly, the reliance on self-reported data from in-person and telephone surveys may introduce bias, as responses can be influenced by respondents’ perceptions and the tendency to provide socially desirable answers [79]. This type of data collection might not accurately capture true behaviors and attitudes. The use of only three suburbs in Ipswich, Queensland, limits the generalizability of the findings to other regions with different socioeconomic and demographic profiles and lower levels of shelter admissions. This might skew the results, as these areas may not represent the broader population’s attitudes and behaviors towards stray cat management. However, the involvement of this targeted population was intentional, and it provides a basis for other areas with a high levels of cat intake into receiving shelters and pounds, as well as high euthanasia levels, to compare their baseline data with our findings. The COVID-19 pandemic necessitated a shift from face-to-face to telephone surveys, also potentially affecting the response rate and the depth of interaction with participants [80]. Finally, while the questionnaire covered a range of relevant topics, the complexity and length of the survey might have led to respondent fatigue, affecting the quality and completeness of the data collected [81].

## 5. Conclusions

This study provides crucial insights into community attitudes and behaviors towards stray cats, highlighting significant challenges and opportunities for effective cat management in urban areas. The situational analysis performed in Ipswich, Queensland, revealed that a majority of residents encounter stray or wandering cats, primarily at private residences, with substantial concerns about predation on native wildlife and nuisance behaviors. These findings underscore the inadequacy of traditional depopulation methods, which have not succeeded in reducing cat-related complaints or impoundments, nor in addressing the community’s priorities for humane and effective cat management.

The data gathered demonstrate a majority of community support for non-lethal management strategies, particularly sterilization-based programs, despite existing legislative barriers in Australia. This preference aligns with international evidence favoring high-intensity trap–neuter–return (TNR) programs, which effectively decrease shelter intake and euthanasia rates, when implemented correctly.

Furthermore, the study highlights the need for providing targeted information for the public regarding the benefits of sterilization programs for cats, including the reduction of wildlife predation, disease transmission, and the mental health impacts on shelter workers involved in euthanasia. By aligning cat management strategies with community preferences and enhancing public awareness, authorities can foster greater support for humane and sustainable solutions, ultimately improving outcomes for both stray cats and urban communities. These outcomes are aligned with the One Welfare philosophy, which describes the interconnectedness of human, animal, and environmental wellbeing.

## Figures and Tables

**Table 1 animals-14-02953-t001:** Location types where stray or wandering cats were seen. Respondents could report more than one location type where they saw stray or wandering cats.

Location Type Where Stray or Wandering Cats Were Seen	Number of Respondents Seeing Stray or Wandering Cats at This Location Type	Percentage of the 237 Respondents Seeing Stray or Wandering Cats at This Location Type ^1^	Percentage of Locations Where Stray or wandering Cats Were Seen (n = 398) ^2^
Private residences	212	89%	53%
Alleyways or streets	90	38%	23%
Suburban parks	34	14%	9%
Vacant blocks or buildings	16	7%	4%
Food outlets	10	4%	3%
Other commercial businesses	10	4%	3%
Community housing	7	3%	2%
Train stations	7	3%	2%
Drains	7	3%	2%
Schools or universities	3	1%	1%
Government buildings	1	0.4%	0.3%
Industrial areas	1	0.4%	0.3%

^1^ A total of 237 respondents reported seeing stray or wandering cats and specified at least one location type. One further respondent reported seeing stray or wandering cats but did not identify any specific location types. ^2^ One respondent identifying two location types would contribute two to this total, one respondent identifying three location types would contribute three to this total, and so on.

**Table 2 animals-14-02953-t002:** Extent to which stray or wandering cats are a problem for the 238 respondents who reported seeing stray or wandering cats in each respondent location—serious = affects my daily life; moderate = regularly affects me; mild = occasionally affects me; no problem = never affects me.

	Serious % (Number)	Moderate % (Number)	Mild % (Number)	No Problem % (Number)	No Response
Where you live	22% (52)	16% (37)	21% (50)	41% (98)	1
Where you work, study, or spend most of your time away from home	7% (16)	4% (9)	6% (15)	83% (197)	1
In local public places	10% (23)	7% (17)	10% (24)	73% (170)	4

**Table 3 animals-14-02953-t003:** Actions taken against stray or wandering cats by the 238 respondents who reported seeing such cats.

Action Category (Number Who Had Taken This Action)	Of the 238 Respondents Who Reported Seeing Stray or Wandering Cats, Percentage That Had Taken This Action ^1^	Percentage of Actions Taken by Respondents against Stray or Wandering Cats (n = 171) ^2^
Chased cats away or used a deterrent (n = 59)	25%	35%
Prevented cats from entering my home (n = 33)	14%	19%
Captured cats to have them removed (n = 19)	8%	11%
Contacted local council (n = 18)	8%	11%
Contacted welfare agency such as the RSPCA (n = 14)	6%	8%
Provided the cats with food (n = 11)	5%	6%
Contacted owner or returned the cat to the owner (n = 7)	3%	4%
Protected cats from harm (n = 3)	1%	2%
Provided sick and injured cats with medical treatment (n = 2)	1%	1%
Desexed the cat (n = 1)	0.4%	0.6%
Other (n = 4)	2%	2%

^1^ Of the 238 respondents who reported seeing stray or wandering cats, 52% (123) reported that they took no action. ^2^ One respondent taking two actions would contribute two to this total, one respondent taking three actions would contribute three to this total, and so on.

**Table 4 animals-14-02953-t004:** Distributions of responses to questions regarding stray cat management strategies shown as % (number). “If two methods of cat management were equally effective in decreasing unowned stray cat numbers over time and the problems they cause, how strongly would you support or approve of the two methods of control for healthy cats that cannot be readily adopted?” They were also then asked, “How strongly do you support just leaving the unowned stray cats alone?” Questions were asked in sequential order, as listed, with each respondent remaining unaware of any additional methods they would subsequently be asked about. Respondents could be supportive (or unsupportive) of all methods.

	Extremely Supportive	Somewhat Supportive	Neutral	Somewhat Unsupportive	Extremely Unsupportive
Catching and humanely euthanizing unowned stray cats that cannot be readily adopted (n = 336) ^1^	33% (111)	23% (76)	15% (50)	13% (44)	17% (57)
Catching, sterilizing, vaccinating, and returning the healthy stray cats to where they live (n = 333) ^2^	31% (103)	21% (71)	15% (49)	11% (38)	22% (72)
How strongly do you support just leaving the unowned stray cats alone? (n = 333) ^2^	7% (22)	9% (31)	15% (51)	17% (56)	52% (173)

^1^ A total of nine study respondents did not respond to this question. ^2^ A total of 12 study respondents did not respond to this question.

**Table 5 animals-14-02953-t005:** Associations between respondents fostering wildlife, feeding or caring for animals not owned by the respondent, and pet ownership and their responses to the question, “If studies showed, that to decrease stray cat numbers in Ipswich, either 3000 stray cats would need to be killed or desexed (sterilized), which method of stray cat management would you prefer?” (% (number)).

Respondent’s Behavior	Support Euthanizing	Support Sterilizing ^1^	Significance of Association between Attribute and Support for Euthanizing
Foster wildlife	11% (1)	89% (8)	*p* = 0.17
Do not foster wildlife	36% (95)	64% (171)	
Regularly feed or care for unowned birds	36% (21)	64% (38)	*p* = 0.88
Do not regularly feed or care for unowned birds	35% (77)	65% (146)	
Regularly feed or care for possums	45% (5)	55% (6)	*p* = 0.52
Do not regularly feed or care for possums	34% (93)	66% (178)	
Regularly feed or care for unowned cats	39% (9)	61% (14)	*p* = 0.65
Do not regularly feed or care for unowned cats	34% (89)	66% (170)	
Own birds	40% (10)	60% (15)	*p* = 0.66
Do not own birds	34% (87)	66% (167)	
Own dogs	34% (50)	66% (98)	*p* = 0.80
Do not own dogs	36% (47)	64% (84)	
Own cats	**25% (26)**	**75% (77)**	***p* = 0.01**
Do not own cats	**40% (71)**	**60% (105)**	

^1^ “Support” = extremely or somewhat supportive; bold text indicates an association with a low *p*-value.

**Table 6 animals-14-02953-t006:** Respondent preferences for council managed programs of unowned stray cats in respondent’s suburb.

	Very Important	Important	Moderately Important	Low Importance	Not at All Important
Decrease killing of birds and small native animals by cats (n = 330) ^1^	72% (238)	19% (61)	6% (21)	3% (9)	0.3% (1)
Stop stray kittens being born (n = 333) ^1^	68% (226)	26% (86)	5% (17)	0.3% (1)	1% (3)
Reduce the risk of disease spread to wildlife (n = 334) ^1^	65% (218)	24% (79)	6% (19)	5% (15)	1% (3)
Reduce the risk of disease spread to pets (n = 331) ^1^	61% (201)	29% (95)	6% (19)	4% (14)	0.6% (2)
Reduce the risk of disease spread to humans (n = 334) ^1^	61% (201)	27% (90)	7% (24)	5% (15)	1% (4)
Stop cats fighting with my cat(s) (n = 152) ^2^	56% (85)	15% (22)	13% (19)	5% (7)	13% (19)
Find homes for stray cats (n = 334) ^1^	47% (158)	37% (122)	9% (30)	3% (10)	4% (14)
Decrease stray cat numbers in my suburb (n = 334) ^1^	46% (154)	28% (95)	14% (48)	7% (22)	5% (15)
Stop nuisance behaviors such as defecating (pooing) and urinating (peeing) in my yard or public places near where I live or work (n = 330) ^1^	43% (143)	26% (84)	12% (39)	11% (37)	8% (27)
Healthy cats and kittens are not euthanized or killed (n = 331) ^1^	41% (137)	21% (69)	18% (59)	10% (33)	10% (33)
Stop cats coming on my property (n = 330) ^1^	38% (125)	20% (67)	16% (52)	12% (39)	14% (47)
Remove the cats causing concern to me (n = 333) ^1^	36% (120)	25% (83)	17% (55)	10% (33)	13% (42)
Be cost effective to councils and rate payers (n = 335) ^1^	32% (108)	27% (91)	28% (92)	5% (15)	9% (29)

^1^ n = number of respondents who recorded a rank for this option. For each option, between 8 and 13 respondents did not respond. ^2^ a total of 171 respondents selected “Not applicable”, and 20 did not respond.

## Data Availability

The most relevant data are reproduced in the text.

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
