# Peer review of "A Situational Analysis of Attitudes toward Stray Cats and Preferences and Priorities for Their Management"

_animals, 2024, doi:10.3390/ani14202953_

Round 1

Reviewer 1 Report

Comments and Suggestions for Authors

Comments on the Quality of English Language

Some comments have been made within the review above.

Author Response

Thank you very much for your very helpful reviewing of our manuscript which has resulted in a more valuable and readable article. We are very grateful to you for taking the time to provide detailed advice on the revision which was very helpful and much appreciated. Please see attached file for detailed responses. Thank you, Jacquie 

Reviewer 2 Report

Comments and Suggestions for Authors

Although there are no major issues with the layout and writing of this work the content is rather basic.

Despite presenting information from a specific situation it lacks depth that could have been provided through a thematic analysis. The references are very focussed and information could be used to support ideas from other contexts, especially regions which have similar issues with cat management and wildlife conflicts.

I feel this manuscript could provide useful context to a wider study if published in a single manuscript but, on its own, doesn't really contribute meaningfully to the wider literature.

Author Response

(The authors gave the same response as above.)

Reviewer 3 Report

Comments and Suggestions for Authors

I read the paper A situational analysis of attitudes to stray cats and preferences and priorities for their management. with interest. It is a nice study looking at the perception of the inhabitants of 3 suburbs in Ipswich on the topic of stray cats, their impacts on society, and management preferences. Similar to other studies, the authors found more support for sterilisation programs than for lethal management of the population. This is really interesting, as, like the authors state, in Australia this is certainly not the method of choice, and is generally very much disregarded as a method that wont work anyway. Which, of course, is contrary to what research from overseas tells us, as these programs can be very effective if properly implemented. Maybe if public support increases, and the social license for lethal control decreases, even Australia could change.

In general, I think it is good research, and the paper well written. I do not have any major comments on it, only some specific ones, detailed below.

General in the paper: why is ‘females’ and ‘males’ used for humans (generally this is more used for animals), and not ‘women’ and ‘men’?

Abstract

Line 29-30. ‘which negatively …. Involved staff’. I am not sure where that is mentioned in the paper, and it is unclear. Which staff?

Line 44. I thought pet ownership was an important predictor, mainly cat ownership?

Lines 26-27 and 46-47. I thought there was already favoured support for sterilisation compared to lethal control? I agree that public education would likely further enhance that support, but as a conclusion, would it not be better to also emphasize the preference for non-lethal to start with?

Introduction

The introduction is good, I would just suggest that the authors make it a little more clear that this paper reports on the other part of the situational analysis. Currently, because they already mention the situational analysis and summarise the results of that which have already been submitted for publication, it gets a little confusing when they then state that this paper’s aim was to conduct a situational analysis. It would be more clear to state that the aim of this paper is to report on the other half of the situational analysis that was conducted, dealing with management preferences, attitudes and priorities.

Materials and methods

Line 126. ‘Data was obtained….Qld’. Ok, but how were respondents selected? Did every individual receive the questionnaire? Was participation voluntary? How many responses were received? This is all very important information.

Line 140. I would say ‘the majority of respondents were female’

Lines 143-150, ok I found the info about respondent selection etc. It might be a little more clear to re-arrange these sections a little, to avoid the confusion I encountered. The first section can only deal with the questionnaire, so the sentence in line 126-128 ‘data were obtained …. Qld’ can be removed from there. This sentence can be added to the following section (now starting at line 136, ‘Suburbs selected…..’), that following section can start with that sentence. That second paragraph then deals with suburb selection and the reasoning behind that. The rest then flows from there.

Lines 149-150. Were these valid responses equally divided between the 3 suburbs?

Line 168-175. ‘respondents were also asked ….. management program’. This section is a little unclear. Did the delay mean that the survey by phone was only conducted after sterilisation had already started? If so, that needs to be more clearly stated, as in this case it is unclear why the difference.

Results

Line 245 ‘(results not shown)’. I am not sure it is necessary to add that there.

Table 5. The p-values are a little lost in this table. It would be more clear to re-arrange in such a way that they do not get lost like that.

Line 306-308. ‘the respondents … very important’. This should be in methods

Discussion

Line 362. ‘and contribute to the ongoing cycle of kittens being born’. That needs a little more elaboration.

Lines 367-374. ‘the council ….. facility (pound)’. This is a good section, but feels like some type of conclusion or recommendation is missing. Do the authors recommend or conclude that more assistive approaches would be beneficial going forward (with regards to cat welfare, costs incurred and shelter worker mental health)? If so, it would be good to elaborate the discussion on this a bit, and state that.

Lines 377. ‘preferring killing’. Delete ‘preferring’.

Line 378. ‘prior research’ needs to be elaborated on a bit.

Line 409-412. ‘this is very similar …. male participants’. For consistency, it is better to either report percentages in the sentence, or in brackets after a general statement, but not use a mix of these in one sentence.

Line 438 and 439. It is unclear what the two percentages in brackets represent.

Lines 434-445. This is a lot of repetition of results. Summarising results is ok to put it in context of what is being discussed, but this whole section seems to consist more of results than of discussion.

Line 460-473. This is a good section, but again I am missing some type of recommendation or conclusion at the end.

Line 486. ‘from online surveys’ I understand from the methods that the data was collected initially in-person and later by phone. If it was also an online survey, this needs to be added to the methods.

Table A4 and A5. The p-values are a bit lost here, it would be better to re-design the layout so the p-value are more clear and do not get lost.

Author Response

(The authors gave the same response as above.)
